# Concurrent Validity and Reliability of a Free Smartphone Application for Evaluation of Jump Height

**DOI:** 10.3390/jfmk9030155

**Published:** 2024-09-01

**Authors:** Amândio Dias, Paulo Pires, Leandro Santana, Paulo Marques, Mário C. Espada, Fernando Santos, Eduardo Jorge Da Silva, André Rebelo, Diogo S. Teixeira

**Affiliations:** 1Egas Moniz Center for Interdisciplinary Research (CiiEM), Egas Moniz School of Health & Science, Caparica, 2829-511 Almada, Portugal; 2Integrative Movement and Networking Systems Laboratory (INMOV-NET LAB), Egas Moniz Center for Interdisciplinary Research (CiiEM), Egas Moniz School of Health & Science, Caparica, 2829-511 Almada, Portugal; 3Sport Physical Activity and Health Research & Innovation Center (SPRINT), 2040-413 Rio Maior, Portugal; mario.espada@ese.ips.pt; 4Centre for the Study of Human Performance (CIPER), Faculdade de Motricidade Humana, Universidade de Lisboa, Cruz Quebrada-Dafundo, 1499-002 Lisboa, Portugal; ppires22@gmail.com; 5Hospital da Luz, 1500-650 Lisboa, Portugal; 6Postgraduate Program in Physical Education, Federal University of Juiz de Fora, Juíz de Fora 36036-900, Brazil; leandrosantana.edufisica@hotmail.com; 7Faculty of Physical Education and Sport, Lusófona University, 1749-024 Lisbon, Portugal; paulojlmfit@gmail.com (P.M.); edujodasilva@gmail.com (E.J.D.S.); p5128@ulusofona.pt (D.S.T.); 8Instituto Politécnico de Setúbal, Escola Superior de Educação, 2914-504 Setúbal, Portugal; fernando.santos@ese.ips.pt; 9Life Quality Research Centre (CIEQV-Leiria), 2040-413 Rio Maior, Portugal; 10Comprehensive Health Research Centre (CHRC), Universidade de Évora, 7004-516 Évora, Portugal; 11Life Quality Research Centre (CIEQV-Setúbal), 2914-504 Setúbal, Portugal; 12Faculdade de Motricidade Humana, Universidade de Lisboa, Cruz Quebrada-Dafundo, 1499-002 Lisboa, Portugal; 13CIDEFES, Centro de Investigação em Desporto, Educação Física e Exercício e Saúde, Universidade Lusófona, 1749-024 Lisbon, Portugal; andre.gomes.rebelo@ulusofona.pt; 14COD, Center of Sports Optimization, Sporting Clube de Portugal, 1600-464 Lisbon, Portugal

**Keywords:** neuromuscular, jump, validation, Android

## Abstract

**Background/Objectives**: Jump test assessment is commonly used for physical tests, with different type of devices used for its evaluation. The purpose of the present study was to examine the validity and reliability of a freely accessible mobile application (VertVision, version 2.0.5) for measuring jump performance. **Methods:** With that intent, thirty-eight college age recreationally active subjects underwent test assessment after a specific warm-up, performing countermovement jumps (CMJs) and squat jumps (SJs) on a contact platform while being recorded with a smartphone camera. Jump height was the criterion variable, with the same formula being used for both methods. Data analysis was performed by two experienced observers. **Results:** The results showed strong correlations with the contact platform (ICC > 0.9) for both jumps. Furthermore, between-observer reliability was also high (ICC > 0.9; CV ≤ 2.19), with lower values for smallest worthwhile change (≤0.23) and typical error of measurement (≤0.14). Estimation error varied when accounting for both observers, with the SJ accounting for bigger differences (4.1–6.03%), when compared to the CMJ (0.73–3.09%). **Conclusions:** The study suggests that VertVision is a suitable and handy method for evaluating jump performance. However, it presents a slight estimation error when compared to the contact platform.

## 1. Introduction

The jump test is widely used in sports as well as research, since is one of the simplest physical tests to address neuromuscular performance in lower limbs [1,2]. In this type of test, two jumps are commonly used, namely, the countermovement jump (CMJ) and squat jump (SJ), with the CMJ reported to have greater performance, due to the effects of the stretch–shortening cycle [3]. Vertical jump tests have also been used to assess both athletic [2,4] and non-athletic populations, such as elderly people [5,6] or children [7].

There are different methods for measuring vertical jump height, with force plates being considered the “gold standard” equipment [1]. Force plates can measure vertical jump height using flight time or take-off velocity methods [8]. However, this type of equipment presents an elevated cost, which creates a substantial barrier for its use among coaches and researchers [9]. Therefore, other methods and types of equipment have been used, such as infrared cells [10], accelerometer-based systems [11], or contact mats [12,13]. These technologies are more inexpensive and allow for assessing athletes in their training setting [14], due to their portability, but are not without constraints, since some tend to have poor agreement and overestimate the measurements [11].

Smartphones combine ease of transportation and handling with a high level of technology and connectivity, making them the ideal device for test assessment in real time, as well as for keeping the stored information for posterior consultation [15]. So, naturally, smartphone applications (apps) have been developed and are gaining space as valid tools for physical test assessment and research. The most researched and tested app is My Jump, which] has been proven valid and reliable for estimating jump height in athletes [16,17], children [18], and the elderly [19]. It was first validated by Balsalobre-Fernández et al. [20] and is available in IOS and Android devices, with both being validated [9,20]. Another app that has been demonstrated to be valid and reliable for assessing jump height is Jumpo, which was first validated by Vieira et al. [21] and posteriorly compared to My Jump [9], which is freely available for android devices. The apps operate in similar manners, by using a time-in-air method to determine jump height. They use the phone’s camera to record the jumps, and afterwards, by manually selecting the take-off and landing frames, flight time is determined. Nevertheless, the method used for both apps is not infallible, since some measurement error can occur when compared to the force platform [20,21].

However, there are other smartphone applications that claim to also assess jump height as well as other physical tests but whose validity and usefulness are questionable [22], since almost none has been tested and compared to other validated equipment and technologies [23]. Unlike the iPhone operative system (iOS, version 17), which runs on only one type of smartphone, the Android operating system runs on multiple devices, with different types of features regarding software and hardware, such as camera resolution. Since these are the most common in market share [24], it is natural that more apps emerge in this mobile environment. One such app is VertVision (VTV), which uses a method for determining jump height similar to that of My Jump or Jumpo but whose validity or reliability is yet to be determined. Therefore, the purpose of the present study was to determine the concurrent validity and reliability of the VTV app, running on the Android system, for measuring jump performance by estimating jump height.

## 2. Materials and Methods

### 2.1. Participants

In order to determine the number of subjects required, a sample size estimation was calculated [25]. For an expected reliability of 0.8 and minimum reliability of 0.6, with 90% power, a two-tailed significance level of 0.05, and an expected drop-out rate of 10%, the number of required subjects was established at 36. For the present study, 38 sports science students, who were recreationally active, agreed to participate (age: 21.84 ± 3.48 years; body mass: 69.24 ± 11.29 kg; height: 1.74 ± 0.09 m). Of these, four were female, and all the others were male. For inclusion in the study, subjects were required to be free of injury and lower-extremity pain at the moment of data collection, as well as in the last three months. Written informed consent was obtained from all participants prior to data collection, explaining the aims and risks of participating in the study. The study was approved by the Ethics Committee of the Polytechnic Institute of Leiria (CE/IPLEIRIA/22/2021) and considered the procedures mentioned in the Declaration of Helsinki [26].

### 2.2. Instruments

The investigation was performed in an academic sports laboratory where jump assessments were conducted using a Xiaomi Mi 11 lite (Xiaomi, Beijing, China) smartphone positioned on a tripod. The tripod was situated 1.5 m [20] from where jumps were performed, at a height of 30 cm [27], ensuring an unobstructed frontal view of each participant’s lower limbs to clearly capture the moments of take-off and landing. Videos were captured at a high frame rate of 240 Hz and a resolution of 720 pixels. Following data collection, all footage was uploaded to a cloud service for subsequent analysis by the observers. All videos were analyzed using a Samsung Galaxy Lite 7 (Samsung, Seoul, Republic of Korea), which had installed version 14.0 of Android. The tablet used for analysis did not have the capacity for recording at 240 Hz, hence the need for using additional equipment for data recording. Measurements from VTV (version 2.0.5) were benchmarked against data from a contact platform (ChronoJump Boscosystem, Barcelona, Spain) which is recognized for its validated accuracy [13], to provide a comparative standard for evaluating the app’s performance.

### 2.3. Design and Procedures

The study was designed as an observational experiment in which all data were captured in a single session. Initial measurements included leg length (greater trochanter to toe tip) and leg height at a 90-degree flexion (vertical line from the greater trochanter to the floor), which was required for jump height estimation in the Chronojump software (version 1.9.0). Participants underwent a standardized warm-up that included dynamic stretching and three practice trials. They executed three SJs and three CMJs, taking a 30-second rest between jumps, with the sequence randomized. No specific instructions were given on the CMJ depth to preserve natural movement, whereas for SJs, participants maintained a squat position at approximately 90 degrees for two seconds before jumping. Subjects were instructed to maintain lower leg extension while jumping (no knee bending) and land with both feet on the contact platform. Any jump that did not fulfill the protocol instructions was redone. Testing conditions such as lighting and sound were consistently maintained, with sessions conducted at the same time of day.

Participants avoided strenuous physical activities 24 h before testing and wore appropriate attire. Safety was ensured by maintaining a clear space around the contact platform, and a whiteboard with unique identifiers was placed behind each participant for easier identification during later video analysis. Experienced evaluators independently assessed a total of 456 jumps, using criteria to select key video frames at the moments of take-off and landing. The order of video analysis was randomized and jump height data were specifically compared. The app provided jump height with the imperial system (inches), which was converted to the metric system (cm) with the following formula: height cm = height inches × 2.54. Jump height for both measurement methods was estimated with the same formula: h = t^2^ × g/8, where *g* is the gravity acceleration (9.81 m/s^2^), h is jump height, and *t* is flight time.

### 2.4. Statistical Procedures

Statistical analysis commenced with the derivation of descriptive statistics, specifically the mean and standard deviation. The normality of the data was verified using the Shapiro–Wilk test. Disparities between measurements taken by different observers and the contact platform were evaluated using paired samples t-tests for both the SJ and CMJ, specifically focusing on jump height. Only the highest scores from the three jumps were considered for these calculations. To quantify the extent of changes, standardized mean differences with 95% confidence intervals and Hedges’s g for effect size (ES) were computed [28], categorizing the ES as trivial if g was <0.2, small if 0.2–0.5, moderate if 0.5–0.8, large if 0.8–1.60, and very large if >1.60 [29]. Reliability assessments included calculations of the Intraclass Correlation Coefficient (ICC) using both single- and multiple-rater models under a two-way random effect scheme [30]. ICC values of <0.5 were considered indicative of poor reliability, values of 0.5–0.75 were indicative of moderate reliability, values of 0.75–0.90 were indicative of good reliability, and values of >0.90 suggested excellent reliability [30]. This analysis was facilitated through Statistical Package for Social Sciences (SPSS, v26, IBM Corp., Armonk, NY, USA). Additionally, Typical Error (TE), expressed as the coefficient of variation (CV), and the smallest worthwhile change (SWC), defined as 0.2 times the between-subject standard deviation, were calculated using an Excel template [31]. High reliability was determined if the ICC was >0.90 and the CV was <5% [32]. The usefulness of the test was defined as “marginal” (TE > SWC), “OK” (TE = SWC), or “good” (TE < SWC). Concordance in measurements was evaluated using Lin’s concordance correlation coefficient (CCC) [33] where values of >0.95 were deemed necessary to consider a good agreement [34].

Additionally, the percentage of estimation error was computed using the following equation [35]: % estimation error = 100 × (A − B)/A, where A is the average CMJ and SJ using the contact platform and B is the average CMJ and SJ using VTV. Potential bias between observers using VTV and the contact platform was assessed using Bland–Altman plots.

## 3. Results

Table 1 presents the descriptive information regarding the participants, which was also used for the contact platform.

Intra-observer and contact platform reliability results for CMJ and SJ height are presented in Table 2. In all cases, the ICC scores were >0.90, indicating good reliability.

Inter-rater reliability scores are presented in Table 3 for the CMJ and SJ. For both the CMJ and SJ, height ICC scores were >0.90 and CV was below 5% in all situations. Significant paired differences were observed between both observers for the CMJ and SJ and in both observers and the platform results for the CMJ and SJ. Effect size (g) results were, however, all trivial (<0.2), except between observer 1 and the contact platform for the CMJ (ES = 0.20). As for usefulness, all results were rated as good. All agreement results for the CMJ and SJ indicate good agreement between observers and between observers and the platform (CCC > 0.95).

With regard to estimation error, for the CMJ, the VTV app presented estimation errors of 3.65% for observer 1 and 0.73% for observer 2. For the SJ, the estimation errors were greater, with 6.03% for observer 1 and 4.19% for observer 2.

The Bland–Altman plots displayed in Figure 1 demonstrate that for both jumps (CMJ and SJ) and both observers, no systematic differences (bias) were present.

## 4. Discussion

The aim of the present study was to analyze the concurrent validity and reliability of the VTV app for measuring jump height. Both the SJ and CMJ were chosen since they are commonly used by researchers and health practitioners [36]. The app was found to be a highly valid and reliable tool for measuring vertical jump height for the CMJ and SJ. Despite the variability that arose from observers having to choose both key frames for jump assessment manually [37], VTV appears to be a valid tool for jump assessment. The validity of VTV (ICC > 0.9) is equivalent to those of other smartphone applications that have been previously tested, such as Jumpo [21] or the most tested app, My Jump [20,36]. This high level of correlation is also equal to those of other systems used to assess jump height, such as photoelectric cells [38], accelerometers [39], or linear position transducers [40]. So, it seems that sports scientists, athletes, or practitioners could benefit from this easy-to-use app and be confident in the measurements of jump height that it provides.

Reliability is also an equal measure of confirmation of the importance an assessment tool can have since it attests for the consistent and repeatable outcomes it can provide. The VTV app demonstrated a between-observer high reliability (ICC > 0.90 and CV < 5%) and good rating (TE < SWC), with small SWCs for both observers and both jumps. In his study with recreationally active athletes for validating the My Jump 2 app, Bogataj et al. [36] used the same TE measurements as this study [31]. The authors reported levels of CV of 7.2% for the SJ and 4.3% for the CMJ, which are higher than reported by our study. The same authors also evaluated SWC, which was in both jump types above one, also demonstrating higher values than in our study. Another study that looked at the validation of the My Jump app with a force platform [41] reported a TE smaller than in our study (0.02). This could be related to the fact that different reference methods were used, since we utilized a contact platform. The same app was validated in another study [42] that used a contact platform as the reference method, which also presented higher values for CV (4.38–6.06) than the VTV app in our study. Another study also assessed the reliability of the My Jump app with different recording heights [27]. The SWC reported between the three observers for the CMJ was higher (>1) than in the present study. The difference in results between the apps may be due to the equations used to estimate height. The My Jump app uses Bosco equations for height estimation [20], while in the VTV app no information is given on the equations used. Veira et al. [21] also used the CMJ and SJ for validation of another app (Jumpo), with a force platform as the reference method. The results presented an ICC equal to that of the VTV and a CV smaller than in our study for the CMJ (1.2) but more elevated (2.6) for the SJ. The difference in these values can be attributed to the different sampling frequencies in the reference methods used, since they are higher (1000 Hz) than the frame rate in the present study or in the other mentioned studies (240 Hz).

The ES in our study was deemed trivial (g < 0.2) for all comparisons between observers and the contact platform. These results are parallel to those in other studies that also demonstrated trivial ESs for the My Jump app [36,43] and the Jumpo app [21]. The equal ESs between all these smartphone applications demonstrate that despite being both valid and reliable for jump height measurement, they can still somehow present limited practical applications when compared to other reference methods, such as force plates or contact platforms. With regard to bias, the Bland–Altman plots suggest that there was no proportional bias between measurement methods for both observers. This means that the difference between the methods is consistent across all values of the measurements.

Regarding the CMJ, the VTV app showed a small estimation error for the two observers, while the estimation for the SJ was more elevated. They are in the same range as or smaller than in previous studies [8,44]. One possible explanation for the differences may be associated with the different devices used for measurement and their sampling frequencies. Our study used a contact platform with a sampling rate of 1000 Hz, while Aragón-Vargas [44] used a sampling frequency of 300 Hz, which could have explained the differences in estimation error, because of the delay in the take-off and landing phases. Our results differ from those of another study [35] that reported a value of 0.78% for the My Jump app. Despite the difference from our study in the reference method, they used a sampling frequency equal to that in our study. So, the reasoning for lower overestimation values may be associated with the methods used for jump height estimation. While both VTV and Chronojump use flight time for jump height assessment, the force platform used by Carlos-Vivas [35] assessed jump height with take-off velocity. Heights calculated with flight time appear to be significantly higher than when calculated with take-off velocity [8], which could help explain the more elevated overestimation values for VTV.

Despite the difference between sampling frequency in VTV and Chronojump, the height values were very similar, as reflected in the elevated ICC values. The results show a good comparison between VTV and the contact platform. However, VTV displays a percentage of overestimation that seems to be more elevated than other smartphone apps. Nonetheless, VTV is available for free for Android devices, like Jumpo, while My Jump is a paid application available for all smartphone operating systems. Therefore, it can be used as a valid alternative for jump assessment, since it removes the cost barrier other types of equipment have, such as force platforms [9].

To the best of our knowledge, this was the first study that attempted to ascertain the validity and reliability of the VTV smartphone app. Although it demonstrated that VTV can be a valid alternative, the present study is not without limitations. We used as a reference method a contact platform, since it is also important to validate smartphone applications with more commonly used field assessment tools [36]. However, the “gold standard” for these kinds of validations is the force platform [1], so the results of our study need additional validation studies that use a force platform as a reference method. Additionally, there seems to be a degree of overestimation that could be relevant in VTV when compared to other smartphone apps. It seems important that future studies look further into comparing method error, in order to ascertain the validity of the smartphone app as a reliable tool. Data analysis was made by experienced observers, but no data exist when the app is used by inexperienced evaluators. It would be interesting to assess repeatability between observers in a future study, to account for measurement error. The present study was performed with data recording at 240 Hz, which is not accessible to all devices. Studies have demonstrated that 120 Hz is sufficient for jump height assessment [20], but using another smartphone app. There is no information on VTV regarding data collection at 120 Hz. Future studies should address this issue. The results of the present study are valid only for young recreational subjects and cannot be extrapolated to other populations. Upcoming studies should also test VTV validation in other types of populations. Lastly, VTV presents the results in the imperial system, which can cause a difficulty in result interpretation, since most research on this topic is done using the metric system.

## 5. Conclusions

The present study suggests that VTV, a smartphone app for measuring jump height, is valid and reliable. Therefore, given its ease of use and availability, it can be considered an alternative for coaches, sports scientists, or recreationally active athletes to evaluate two of the most commonly used physical tests, the CMJ and SJ.

## Figures and Tables

**Figure 1 jfmk-09-00155-f001:**
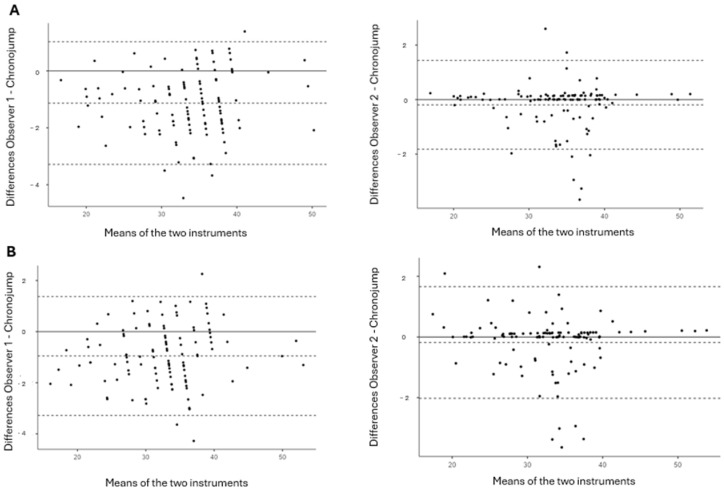
Bland–Altman plots comparing the contact platform with VertVision for both observers with (**A**) the countermovement jump and (**B**) the squat jump. The middle continuous line shows the absolute average difference between devices, with the upper and lower lines showing ±1.96 SD.

**Table 1 jfmk-09-00155-t001:** Descriptive statistics of participants and performed measurements.

Variables	Total Subjects	Male Subjects	Female Subjects
Age (years)	21.84 ± 3.48	21.88 ± 3.51	21.50 ± 3.20
Weight (kg)	69.24 ± 11.29	71.47 ± 9.58	50.25 ± 5.31
Height (m)	1.73 ± 0.08	1.75 ± 0.07	1.58 ± 0.03
Leg length (cm)	102.68 ± 16.27	103.03 ± 17.06	99.75 ± 5.72
Height at 90° flexion (cm)	63.53 ± 6.50	63.59 ± 6.66	63.00 ± 4.85

Note: Values are expressed as mean ± SD.

**Table 2 jfmk-09-00155-t002:** Intra-observer and contact platform reliability for countermovement jump and squat jump for the VertVision.

Variables	Vertvision				Contact Platform	
	Observer 1		Observer 2			
	Mean ± SD	ICC (95% CI)	Mean ± SD	ICC (95% CI)	Mean ± SD	ICC (95% CI)
Countermovement jump						
Jump 1 (cm)	32.737 ± 6.432		33.534 ± 6.292		33.782 ± 6.423	
Jump 2 (cm)	32.676 ± 5.941	0.964 (0.939; 0.980)	33.637 ± 5.888	0.967 (0.944; 0.982)	33.777 ± 5.916	0.969 (0.946; 0.983)
Jump 3 (cm)	32.916 ± 6.186		33.970 ± 6.068		34.164 ± 6.010	
Squat jump						
Jump 1 (cm)	31.378 ± 6.460		32.376 ± 6.113		32.275 ± 6.244	
Jump 2 (cm)	30.783 ± 8.297	0.908 (0.839; 0.950)	31.386 ± 8.241	0.900 (0.825; 0.945)	32.758 ± 6.397	0.975 (0.947; 0.987)
Jump 3 (cm)	33.160 ± 6.853		33.854 ± 6.766		33.996 ± 6.884	

**Table 3 jfmk-09-00155-t003:** Inter-reliability for Countermovement Jump and Squat Jump.

	Countermovement Jump Height (cm)			Squat Jump Height (cm)		
	Obs. 1 vs. Obs. 2	Obs. 1 vs. Platform	Obs. 2 vs. Platform	Obs. 1 vs. Obs. 2	Obs. 1 vs. Platform	Obs. 2 vs. Platform
Paired diff. (cm) (95% CI)	−1.05 (−1.44; −0.67) *	−1.25 (−1.61; 0.88) *	−0.19 (−0.39;0.002)	−0.69 (−1.08; −0.31) *	−0.84 (−1.21; 0.46) *	−0.14 (−0.44; 0.15)
Paired ES (g)	0.17	0.20	0.03	0.10	0.12	0.02
ICC (95% CI)	0.98 (0.90; 0.99)	0.98 (0.79; 0.995)	0.997 (0.995;0.999)	0.99 (0.97; 0.996)	0.99 (0.96; 0.996)	0.996 (0.99; 0.998)
CCC (95% CI)	0.97 (0.94; 0.98)	0.96 (0.93; 0.98)	0.99 (0.99; 0.997)	0.98 (0.96; 0.99)	0.98 (0.96; 0.99)	0.99 (0.98; 0.995)
TE (95% CI)	0.14 (0.11; 0.18)	0.13 (0.11; 0.17)	0.07 (0.06; 0.09)	0.12 (0.10; 0.16)	0.12 (0.10; 0.15)	0.09 (0.08; 0.12)
CV (%)	1.85	2.19	0.55	1.65	1.79	0.75
SWC (cm)	0.23	0.22	0.12	0.23	0.22	0.18
Rating	Good	Good	Good	Good	Good	Good

* *p* < 0.05; ES: effect size; CI: confidence interval; ICC: intraclass correlation coefficient; CCC: concordance correlation coefficient; TE: typical error; CV: coefficient of variation; SWC: smallest worthwhile change.

## Data Availability

The data presented in this study are available on request from the corresponding author due to privacy issues regarding the participants in the study.

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
