# Peer review of "Concurrent Validity and Reliability of a Free Smartphone Application for Evaluation of Jump Height"

_jfmk, 2024, doi:10.3390/jfmk9030155_

Round 1

Reviewer 1 Report

Comments and Suggestions for Authors

Dear corresponding Author,

thanks for submitting your paper. I read it with high interest and I thank you for letting me know the VertVision app.

Tho be honest the paper is not a very innovative and original one, but at the same time it is well conducted and it is important to have this information for professionals and researchers. 

I want to share with you some comments in order to improve the paper:

1) Please add an appendix at the end of the paper with the links to download and install the app. I verified and it is not present on the Play store. I found it on the web.

2) Paragraph 2.2: I don't understand why you recorded the video of the jumps with a device and then you analyze them on another one. Maybe because the tablet did not record at 240 fps? If yes please write it.

3) good choice to record the videoa at 240 fps. It would be interesting if you recorded two videos of the same jump. One at 120 fps and another one at 240. It would have been useful to understand if 120 con be considered sufficient or not to measure properly the jump height. Please add this information in limitations section

4) I don't understand what is "height at a 90-116 degree flexion", flexion of what? Please be more explicit and modify it along the whole text.

5) According to the data in the table 2 I don't understand why the app showed an overestimation, specially for observer 1. The average values in table 2 for the observer 1 are always smaller respect the measurements of the contact platform. Please can you verify and explain what you mean?

Author Response

Comments 1: Dear corresponding Author,

thanks for submitting your paper. I read it with high interest and I thank you for letting me know the VertVision app.

Tho be honest the paper is not a very innovative and original one, but at the same time it is well conducted and it is important to have this information for professionals and researchers. 

Response 1: Thank you for your consideration and valuable comments.

Comments 2: I want to share with you some comments in order to improve the paper:

1) Please add an appendix at the end of the paper with the links to download and install the app. I verified and it is not present on the Play store. I found it on the web.

Response 2: Thank you for your comment. Regrettably, you are correct, in the time between the present study being conducted and the manuscript submission, the smartphone application was removed from Google play store, albeit it is still accessible through multiple sources found online. We have added two of them in the supplementary materials.

Comments 3: 2) Paragraph 2.2: I don't understand why you recorded the video of the jumps with a device and then you analyze them on another one. Maybe because the tablet did not record at 240 fps? If yes please write it.

Response 2: Thank you for your comment. The tablet used for analysis did not have the capacity for recording at 240 Hz, hence the need for using additional equipment for data recording. This information was added on topic 2.2 Instruments.

Comments 3: 3) good choice to record the video at 240 fps. It would be interesting if you recorded two videos of the same jump. One at 120 fps and another one at 240. It would have been useful to understand if 120 con be considered sufficient or not to measure properly the jump height. Please add this information in limitations section

Response 3: Thank you for your comment. Other studies acknowledge that 120fps is enough for jump height assessment, but they were performed on a different smartphone application. This limitation has been included in the manuscript, as mentioned

Comments 4: 4) I don't understand what is "height at a 90-116 degree flexion", flexion of what? Please be more explicit and modify it along the whole text.

Response 4: Thank you for your comment.  You must be referring to the sentence “. Initial measurements included leg length and height at a 90-degree flexion to facilitate(…)”, which was on line 116, at the beginning of topic 2.3 Design and procedures. The line ended on “90-“ and afterward is the line number. Maybe there was some kind of formatting error.

For jump assessment, Chronojump software requires 2 measurements. Leg length: greater trochanter to toe-tip and leg height at a 90-degree flexion: vertical line from the greater trochanter to the floor. This information has been added to the manuscript.

Comments 5: 5) According to the data in the table 2 I don't understand why the app showed an overestimation, specially for observer 1. The average values in table 2 for the observer 1 are always smaller respect the measurements of the contact platform. Please can you verify and explain what you mean?

Response 5: Thank you for your comment. When we refer to overestimation, we are mentioning the estimation error. We comprehend that it can cause an inaccurate assumption and have changed the term to estimation error throughout the manuscript.

Reviewer 2 Report

Comments and Suggestions for Authors

The study compared the jump height calculated using a contact platform and a video recorded by a smartphone, using time in both cases. Since the assessments were made simultaneously, this means that the results should differ only due to:

1. The measurement frequency of the devices - 1000 Hz for conact platform and 240 Hz for video recording by smartfon. Notice the significant difference in the frequency registration. Does it not affect the final result?

2. The method of determining the time of foot take-of from the ground and contact during landing - the contact platform has a mechanical mechanism set by the manufacturer, and the error in assessing the time of take-off and landing in the video recording depends on the operator.

Since experienced evaluators independently assessed a total of 456 jumps, using 128 criteria to select key video frames at the moments of take-off and landing, the results of this assessment were probably repeatable. However, they cannot be transferred to practice because take-off and landing points will not be determined by experienced evaluators. What error will occur when assessing the jump height by a person who has no experience? The article aims to examine the similarity between the two methods. However, the authors do not describe the errors in the method used. Determining the points of contact and foot take-off will be more accurate for the higher frequency of recorded images. Moreover, in each video analysis, the quality of the analysis depends on the positioning of the camera relative to the object.

Despite everything, both methods unfortunately have a measurement error, which is an error resulting from incorrectly assumed initial and final conditions. After all, a person performing a jump does not have to land on almost straight limbs - they can bend them and the degree of bending will increase the height of the jump in this method. This will lead to false conclusions, which are not found in measurements using force platforms.

Other comments.

Line 163 - what were the values ​​of Leg length (cm) and Height at 90º flexion (cm) used for - there no description anywhere? Why are you dividing the group by gender?

Line 167 - Table 2. Intra should be Table 2. Intra

General comments:

1. I suggest not writing that a smartphone app for measuring jump height is an important and reliable tool for young recreationally active people - because it requires training of the person performing the measurement. However, it is not suitable for use in scientific research.

2. I suggest supplementing the description of errors in both measurement methods in the introduction - both are not perfect - why?

3. It would be more interesting to determine the jump height by experienced evaluators and non-experienced evaluators. Then you could assess the actual repeatability of this method and talk about its practical use.

4. Add the Bland–Altman plots to estimate the bias.

5. You need to know what formulas the app uses to calculate the height in order to compare both tools. Correct estimation of the maximum height in a vertical jump depends on an accurate and reliable measurement tool. You cannot describe the results without knowing the tool.

6. The calculation of the vertical jump based on the flight time is also provided by the Optojump photoelectric cells and high-speed cameras with the e.g. Kinovea software and many others. Assess who you would like to target with this smartphone-based software. You cannot target it to people involved in competitive sports because there changes of 2% can be significant and the proposed method probably has a larger error - what? Assess the error of the method, not its repeatability.

Author Response

Comments 1: The study compared the jump height calculated using a contact platform and a video recorded by a smartphone, using time in both cases. Since the assessments were made simultaneously, this means that the results should differ only due to:

  1. The measurement frequency of the devices - 1000 Hz for contact platform and 240 Hz for video recording by smartphone. Notice the significant difference in the frequency registration. Does it not affect the final result?

Response 1: Thank you for your comment. Indeed, it is possible for the different equipment used, which records data at different frequencies to affect data outcome. That is why we present the estimation error (previously mentioned as overestimation in the 1st version of the manuscript), so that practitioners, coaches, and athletes could be aware of such differences. Several studies have used similar methods while assessing other smartphone applications, such as:

Tallis J, Morris RO, Duncan MJ, Eyre ELJ, Guimaraes-Ferreira L. Agreement between Force Platform and Smartphone Application-Derived Measures of Vertical Jump Height in Youth Grassroots Soccer Players. Sports (Basel). 2023 Jun 13;11(6):117. doi: 10.3390/sports11060117.

Bishop C, Jarvis P, Turner A, Balsalobre-Fernandez C. Validity and Reliability of Strategy Metrics to Assess Countermovement Jump Performance using the Newly Developed My Jump Lab Smartphone Application. J Hum Kinet. 2022 Sep 8;83:185-195. doi: 10.2478/hukin-2022-0098.

Comments 2: 2. The method of determining the time of foot take-of from the ground and contact during landing - the contact platform has a mechanical mechanism set by the manufacturer, and the error in assessing the time of take-off and landing in the video recording depends on the operator.

Since experienced evaluators independently assessed a total of 456 jumps, using 128 criteria to select key video frames at the moments of take-off and landing, the results of this assessment were probably repeatable. However, they cannot be transferred to practice because take-off and landing points will not be determined by experienced evaluators. What error will occur when assessing the jump height by a person who has no experience? The article aims to examine the similarity between the two methods. However, the authors do not describe the errors in the method used. Determining the points of contact and foot take-off will be more accurate for the higher frequency of recorded images. Moreover, in each video analysis, the quality of the analysis depends on the positioning of the camera relative to the object.

Response 2: Thank you for your comment. Concerning the keyframe selection method used (take-off and landing), it is similar to those previously used for jump height determination in two different smartphone applications validations (Vieira A, Blazevich AJ, DA Costa AS, Tufano JJ, Bottaro M. Validity and Test-retest Reliability of the Jumpo App for Jump Performance Measurement. Int J Exerc Sci. 2021 Aug 1;14(7):677-686.; Haynes T, Bishop C, Antrobus M, Brazier J. The validity and reliability of the My Jump 2 app for measuring the reactive strength index and drop jump performance. J Sports Med Phys Fitness. 2019 Feb;59(2):253-258. doi: 10.23736/S0022-4707.18.08195-1).

The use of such method by inexperienced evaluators has been deemed valid and reliable previously (Balsalobre-Fernández C, Glaister M, Lockey RA. The validity and reliability of an iPhone app for measuring vertical jump performance. J Sports Sci. 2015;33(15):1574-9. doi: 10.1080/02640414.2014.996184), therefore the error that may occur is not different than those that have expertise.

Regarding the errors between methods, as mentioned in the previous answer we used estimation error (previously mentioned overestimation in the 1st version of the manuscript), for assessing differences between devices used.  

You are correct when you state that “Determining the points of contact and foot take-off will be more accurate for the higher frequency of recorded images”, and that is the reason why we have used 240 Hz for video collection and not a lower recording frequency.

With regards to the positioning of the camera, it was always immobile and, in a setup, previously used (Balsalobre-Fernández C, Glaister M, Lockey RA. The validity and reliability of an iPhone app for measuring vertical jump performance. J Sports Sci. 2015;33(15):1574-9. doi: 10.1080/02640414.2014.996184). Additionally, an earlier study has determined that the changes in smartphone height position did no alter significantly the jump height estimation while using the key frame selection method (Jimenez-Olmedo JM, Pueo B, Mossi JM, Villalon-Gasch L. Reliability of My Jump 2 Derived from Crouching and Standing Observation Heights. Int J Environ Res Public Health. 2022 Aug 10;19(16):9854. doi: 10.3390/ijerph19169854). So, quality of the analysis does not depend on the positioning of the camera, as long as it is on the same plane as the jump, which was the case in our study.

 Comment 3: Despite everything, both methods unfortunately have a measurement error, which is an error resulting from incorrectly assumed initial and final conditions. After all, a person performing a jump does not have to land on almost straight limbs - they can bend them, and the degree of bending will increase the height of the jump in this method. This will lead to false conclusions, which are not found in measurements using force platforms.

Response 3: Thank you for your comment.

You are correct and that is why all subjects were instructed to not bend the knees while in the air and land with both feet on the contact platform. If the jump did not meet these requirements, it would be redone. We have added this information on topic 2.3 Design and procedures

Comments 4: Other comments.

Line 163 - what were the values of Leg length (cm) and Height at 90º flexion (cm) used for - there no description anywhere? Why are you dividing the group by gender?

Response 4: Thank you for your comment. Those measurements are a requirement of the software used (Chronojump) for jump estimation. We have added the explication on the manuscript. With regards to gender, the data presented in Table 1 are just for the characterization of the subjects. Data analysis was done on all the data and not divided by gender.

Comment 5: Line 167 - Table 2. Intra should be Table 2. Intra

Response 5: Thank you for your comment. We have made the correction.

Comments 6: General comments: 1. I suggest not writing that a smartphone app for measuring jump height is an important and reliable tool for young recreationally active people - because it requires training of the person performing the measurement. However, it is not suitable for use in scientific research.

 Response 6:  Thank you for your comment. We have changed the conclusions section accordingly.

Comments 7: 2. I suggest supplementing the description of errors in both measurement methods in the introduction - both are not perfect - why?

Response 7: Thank you for your comment, we have added the information in the introduction section.

Comments 8: 3. It would be more interesting to determine the jump height by experienced evaluators and non-experienced evaluators. Then you could assess the actual repeatability of this method and talk about its practical use.

Response 8: Thank you for your comment. Indeed, it would be a very interesting study, and we have added the suggestion in the manuscript.

 Comments 9: 4. Add the Bland–Altman plots to estimate the bias.

Response 9: Thank you for your comment. The Bland-Altman plots for both observers and both types of jumps have been added to the manuscript.

 Comments 10: 5. You need to know what formulas the app uses to calculate the height in order to compare both tools. Correct estimation of the maximum height in a vertical jump depends on an accurate and reliable measurement tool. You cannot describe the results without knowing the tool.

Response 10: Thank you for your comment. The formula used to calculate jump height was added to the design and procedures section.

Comments 11: 6. The calculation of the vertical jump based on the flight time is also provided by the Optojump photoelectric cells and high-speed cameras with the e.g. Kinovea software and many others. Assess who you would like to target with this smartphone-based software. You cannot target it to people involved in competitive sports because there changes of 2% can be significant and the proposed method probably has a larger error - what? Assess the error of the method, not its repeatability.

Response 11: Thank you for your comment. The use of a smartphone high-speed camera for jump height estimation has shown to be a valid method, with several manuscripts published. Despite this, few manuscripts quantify the measurement error between methods. We have tried to do that, by calculating estimation error, but we acknowledge that it may not be the best method. We have added this information in the discussion section as a limitation of the study.

Reviewer 3 Report

Comments and Suggestions for Authors

First of all, I would like to thank the publisher, the journal and the editors for their confidence in reviewing this manuscript in their journal.

The article presents the validation of another possible application for the measurement of jump height. It is an article that fits in with the editorial line of this special issue, but here are some suggestions that the authors can follow to improve the article and make it suitable for publication:

1) In the abstract, the methodology used should be better specified.

2) In the introduction, when explaining various applications that deal with the measurement of jumping, it would be advisable to cite more research validating these applications, using the references of the authors who created the application.

3) In the methodology, I find one of the main problems. Firstly, the sample. The sample, in my opinion, is too small to show the validity of an instrument. Furthermore, showing differences between men and women, comparing 34 men and 4 women, does not seem relevant to me, nor do I think it is a variable that should be taken into account. Was any other variable taken into account that could give more weight to the work? 

4) The results and discussion will be affected by the changes or justification to be made in the methodology section.

5) Finally, a thorough review of all bibliographical references is recommended, as many of them contain errors.

Author Response

Comments 1: First of all, I would like to thank the publisher, the journal and the editors for their confidence in reviewing this manuscript in their journal.

The article presents the validation of another possible application for the measurement of jump height. It is an article that fits in with the editorial line of this special issue, but here are some suggestions that the authors can follow to improve the article and make it suitable for publication:

1) In the abstract, the methodology used should be better specified.

Response 1: Thank you for your comment. We have added some more information about the methodology used in the abstract. However, the word limitation (200 words) prevents us from a thorough description of the methods used.

Comments 2: 2) In the introduction, when explaining various applications that deal with the measurement of jumping, it would be advisable to cite more research validating these applications, using the references of the authors who created the application.

Response 2: Thank you for your comment. The references for the applications were already in the manuscript, but there was no direct mention to them. We have changed the introduction section so that an explicit reference is present for the smartphone applications.

Comments 3: 3) In the methodology, I find one of the main problems. Firstly, the sample. The sample, in my opinion, is too small to show the validity of an instrument. Furthermore, showing differences between men and women, comparing 34 men and 4 women, does not seem relevant to me, nor do I think it is a variable that should be taken into account. Was any other variable taken into account that could give more weight to the work? 

Response 3: Thank you for your comment. As stated in the participants section, we did a pre-study sample size estimation. For an expected reliability of 0.8 and minimum reliability of 0.6, with 90% power, a two-tailed significance level of 0.05, and an expected drop-out rate of 10%, the number of required subjects was established at thirty-six. For our study we recruited thirty-eight subjects, to ensure that we would meet the requirements in sample size estimation.

Additionally, sample size is equivalent to recent studies on this topic, such as:

Åžentürk D, Yüksel O, Akyildiz Z. The concurrent validity and reliability of the My Jump Lab smartphone app for the real-time measurement of vertical jump performance. Proceedings of the Institution of Mechanical Engineers, Part P: Journal of Sports Engineering and Technology. 2024;0(0). doi:10.1177/17543371241246439

Stojiljković N, Stanković D, Pelemiš V, ÄŒokorilo N, Olanescu M, Peris M, Suciu A, Plesa A. Validity and reliability of the My Jump 2 app for detecting interlimb asymmetry in young female basketball players. Front Sports Act Living. 2024 Apr 4;6:1362646. doi: 10.3389/fspor.2024.1362646.

With regards to gender, the data presented in Table 1 are just for the characterization of the subjects. Data analysis was done on all the data and not divided by gender. The only variable assessed was jump height.

Comments 4: 4) The results and discussion will be affected by the changes or justification to be made in the methodology section.

Response 4: Thank you for your comment. We have made changes to the manuscript according to the given justifications.

Comments 5. 5) Finally, a thorough review of all bibliographical references is recommended, as many of them contain errors.

Response 5: Thank you for your comment. We have made a review on bibliographical references, to correct errors and comply with the instructions to authors guidelines.

Round 2

Reviewer 2 Report

Comments and Suggestions for Authors

All changes were made according to the comments.

Reviewer 3 Report

Comments and Suggestions for Authors

The authors have made the suggested changes and the manuscript could be published as is.